# A complex of BRCA2 and PP2A-B56 is required for DNA repair by homologous recombination

Sara M. Ambjørn [1,2], Julien P. Duxin[1], Emil P. T. Hertz[1], Isha Nasa[3], Joana Duro [1], Thomas Kruse[1], Blanca Lopez-Mendez [1], Beata Rymarczyk[4], Lauren E. Cressey[3], Thomas van Overeem Hansen[5], Arminja N. Kettenbach [3], Vibe H. Oestergaard [2✉], Michael Lisby [2,6✉] & Jakob Nilsson [1✉]

Mutations in the tumour suppressor gene *BRCA2* are associated with predisposition to breast and ovarian cancers. BRCA2 has a central role in maintaining genome integrity by facilitating the repair of toxic DNA double-strand breaks (DSBs) by homologous recombination (HR). BRCA2 acts by controlling RAD51 nucleoprotein filament formation on resected single-stranded DNA, but how BRCA2 activity is regulated during HR is not fully understood. Here, we delineate a pathway where ATM and ATR kinases phosphorylate a highly conserved region in BRCA2 in response to DSBs. These phosphorylations stimulate the binding of the protein phosphatase PP2A-B56 to BRCA2 through a conserved binding motif. We show that the phosphorylation-dependent formation of the BRCA2-PP2A-B56 complex is required for efficient RAD51 filament formation at sites of DNA damage and HR-mediated DNA repair. Moreover, we find that several cancer-associated mutations in BRCA2 deregulate the BRCA2-PP2A-B56 interaction and sensitize cells to PARP inhibition. Collectively, our work uncovers PP2A-B56 as a positive regulator of BRCA2 function in HR with clinical implications for BRCA2 and PP2A-B56 mutated cancers.

[1] The Novo Nordisk Foundation Center for Protein Research, Faculty of Health and Medical Sciences, University of Copenhagen, 2200 Copenhagen, Denmark. [2] Department of Biology, University of Copenhagen, 2200 Copenhagen, Denmark. [3] Biochemistry and Cell Biology, Geisel School of Medicine at Dartmouth College, Dartmouth, NH, USA. [4] Department of Biology, University of Konstanz, 78457 Konstanz, Germany. [5] Department of Clinical Genetics, Copenhagen University Hospital, Rigshospitalet, 2100 Copenhagen, Denmark. [6] Center for Chromosome Stability, Department of Cellular and Molecular Medicine, University of Copenhagen, 2200 Copenhagen, Denmark. ✉email: vibe@bio.ku.dk; mlisby@bio.ku.dk; jakob.nilsson@cpr.ku.dk

Homologous recombination (HR) is an essential cellular process that repairs severe DNA lesions such as DNA double-strand breaks (DSBs) to ensure genome integrity[1]. Women inheriting monoallelic deleterious mutations in the central HR components BRCA1 and BRCA2 are highly predisposed to breast and ovarian cancers[2,3]. HR-mediated repair takes place during S and G2 phases of the cell cycle and uses a homologous DNA sequence, most often the sister chromatid, as a template to repair DSBs in a high-fidelity manner[1].

BRCA2 plays a central role in HR by controlling the formation of RAD51 nucleoprotein filaments on resected RPA-coated single-stranded DNA ends, which can then search for and invade a homologous repair template[4–6]. BRCA2 binds monomeric RAD51 through eight central BRC repeats[7–9] and binds and stabilizes RAD51 filaments through a C-terminal domain[10,11]. An N-terminal PALB2 interaction domain recruits BRCA2 to sites of DNA damage as part of the BRCA1-PALB2-BRCA2 complex[12].

HR is a highly regulated process yet many aspects of its regulation are not fully understood[13]. Phosphorylation of BRCA2 and other HR components by DNA damage kinases (ATM/ATR) and cyclin-dependent kinases has been shown to play a role[1,13–15]. In contrast, a direct role of protein phosphatases in HR is less clear in part due to a lack of understanding of how protein phosphatases recognize their substrates[16–18]. Recent discoveries of consensus binding motifs for protein phosphatases[19–21] now allows for precise dissection of their roles in DNA repair processes.

## Results

### BRCA2 binds PP2A-B56 through a conserved LxxIxE motif.
We previously identified a putative binding motif for the serine/threonine protein phosphatase PP2A-B56 in BRCA2, which is of unknown significance[20]. PP2A-B56 is a trimeric complex consisting of a scaffolding subunit (PPP2R1A-B), a catalytic subunit (PPP2CA-B), and a regulatory subunit of the B56 family (isoforms α, β, γ, δ, and ε). PP2A-B56 achieves specificity by binding to LxxIxE motifs in substrates or substrate specifiers through a conserved binding pocket present in all isoforms of B56[20,22–24] (Fig. 1a–b). The LxxIxE motif in BRCA2 is embedded in a hitherto uncharacterized region between BRC repeat 1 and 2 comprising residues 1102–1132, which is highly conserved spanning more than 450 million years of evolution (190 full-length vertebrate BRCA2 protein sequences analyzed by Clustal Omega multiple sequence alignment) (Fig.1b and Supplementary Data 1). To further explore this binding site, we first validated the interaction in human cells, focusing on the main nuclear isoform of B56, B56γ[25]. In HeLa cells, Myc-tagged fragments of BRCA2 spanning BRC repeats 1 and 2 (Myc-BRCA2[1001–1255]) co-purified with Venus-B56γ (Fig. 1c) (Venus is a variant of YFP). Reciprocally, all components of the trimeric PP2A-B56 complex co-purified with Venus-BRCA2[1001–1255] (Supplementary Fig. 1a, Supplementary Data 2). Additionally, BRCA2 co-purified with both B56α/β and B56γ in *Xenopus* egg extracts (Fig. 1d), consistent with an evolutionarily conserved interaction. Mutation of two of the central residues of the LxxIxE motif, L1114 and I1117, to alanines (referred to as the 2A mutant, Fig. 1b) abrogated the interaction to Venus-B56γ (Fig. 1c), showing that the interaction depends on the LxxIxE motif. The direct and LxxIxE motif-dependent interaction between BRCA2 and B56 was confirmed in vitro by isothermal titration calorimetry (ITC) (Fig. 1e, Supplementary Fig. 1b) and gel filtration chromatography (Supplementary Fig. 1c). The $K_D$ is low micromolar, which might explain why the interaction has not been reported previously. Consistent with our binding data, we detected BRCA2 and the BRCA1-PALB2-BRCA2 complex

partner BRCA1 in proximity to B56γ in HeLa Flp-In T-REx cells using a biotin proximity labelling approach with TurboID[26]-tagged B56γ coupled to mass spectrometry (Supplementary Fig. 1d, Supplementary Data 2).

To determine if BRCA2 could recruit PP2A-B56 to DSBs, we exploited the *Xenopus* egg extract system that allows direct monitoring of proteins binding to DSBs. Either closed circular or linearized DSB-containing plasmids were added to *Xenopus* egg extracts, and proteins co-purifying with the DNA were analyzed by western blotting following plasmid pulldown. We found that B56γ was enriched on DSB-containing plasmid DNA, and that immunodepletion of BRCA2 from the extracts diminished the recruitment of B56γ to the same damaged plasmid (Fig. 1f). Taken together, our results show that BRCA2 binds PP2A-B56 through a highly conserved LxxIxE motif and recruits it to DSBs.

### The BRCA2-PP2A-B56 complex is required for DNA repair by HR.
We next asked whether the interaction between BRCA2 and PP2A-B56 is required for the function of BRCA2 in DNA repair by HR. To address this, we constructed an RNAi knockdown and complementation setup in HeLa DR-GFP Flp-In cells[27] and U2OS Flp-In T-REx cells. This setup allowed transient depletion of endogenous BRCA2 using siRNA-mediated knockdown and complementation with stably expressed siRNA-resistant cDNA constructs of mCherry-MBP- or Venus-MBP-tagged full-length BRCA2 WT or 2A (referred to as BRCA2 WT and 2A). Efficient depletion of endogenous BRCA2 and similar expression levels and chromatin association of the complementation constructs were confirmed by western blot analysis (Supplementary Fig. 2a-c). We then utilized the DR-GFP reporter assay[28] (Fig. 2a) to assess HR-mediated DSB repair. Strikingly, complementation with BRCA2 WT but not BRCA2 2A suppressed the loss of HR-mediated repair resulting from BRCA2 depletion (Fig. 2a, Supplementary Fig. 2d), suggesting that PP2A-B56 binding is required for the function of BRCA2 in HR. Consistent with this result, we found that expression of a genetically encoded inhibitor of PP2A-B56 binding to LxxIxE motifs similarly diminished HR-mediated repair in the DR-GFP reporter assay[28,29] (Supplementary Fig. 2e–f).

BRCA2 is considered essential in most contexts at least in part due to its function in HR and its deletion or depletion leads to lethality[30–34]. To assess the importance of the BRCA2-B56 interaction for cell viability, we performed colony formation assays and determined plating efficiencies for BRCA2 WT and 2A complemented U2OS Flp-In T-REx cells (Fig. 2b). Consistent with the results for HR-mediated repair, expression of BRCA2 WT but not BRCA2 2A suppressed the diminished viability resulting from BRCA2 depletion (Fig. 2b).

Due to impaired DNA repair, loss of BRCA2 function causes hypersensitivity to various DNA damaging agents including DNA interstrand crosslinking (ICL) agents[35], topoisomerase I inhibitors[36], and poly-(ADP-ribose) polymerase (PARP) inhibitors[37,38], which is exploited therapeutically[39]. Accordingly, BRCA2 depletion resulted in hypersensitivity to Mitomycin C (MMC), camptothecin (CPT), and Olaparib (Fig. 2c–e). Consistent with a role for the BRCA2-PP2A-B56 complex in DNA repair, BRCA2 2A expressing cells were significantly more sensitive to these DNA damaging agents than BRCA2 WT expressing cells (Fig. 2c-e).

To investigate the mechanistic basis for the impaired DNA repair in BRCA2 mutant cells, we looked at MMC-induced nuclear RAD51 repair foci by immunofluorescence microscopy. BRCA2 depletion abolished the ability to form RAD51 foci (Fig. 2f, Supplementary Fig. 2g), consistent with the central role of BRCA2 in forming RAD51 filaments at sites of DNA damage[35,40].

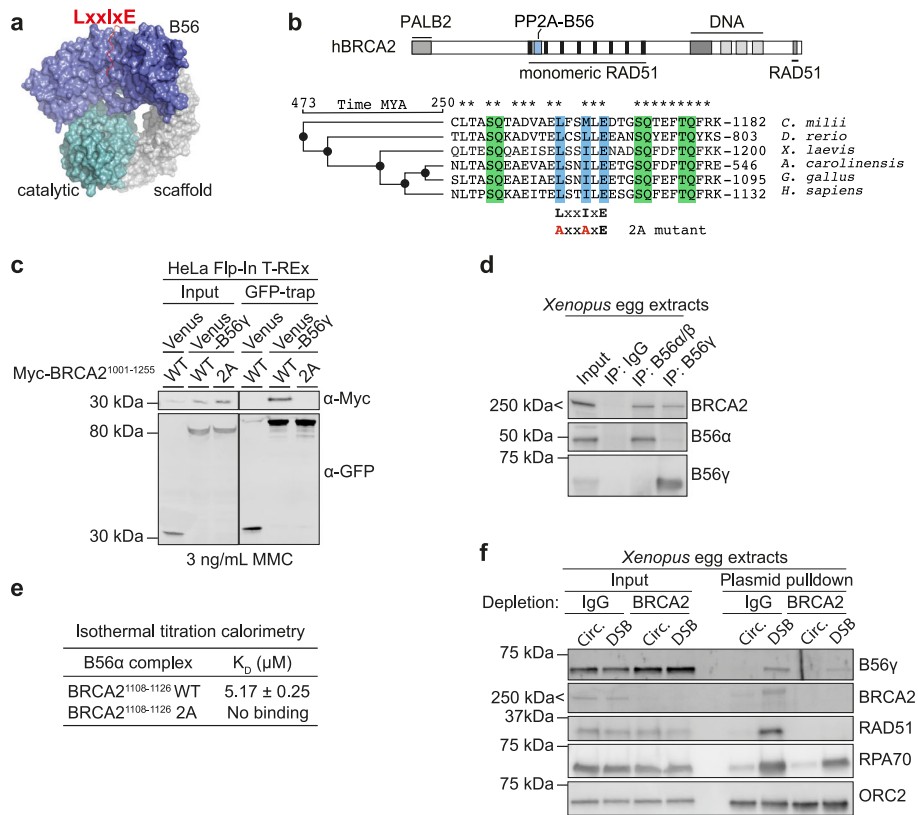

**Fig. 1 BRCA2 binds PP2A-B56 through a conserved LxxIxE motif and recruits it to DSBs. a** Model of the PP2A-B56 holoenzyme with an LxxIxE motif-containing peptide bound generated using PyMOL. **b** Top: Domain organization of human BRCA2 with selected interaction domains and the PP2A-B56 binding region indicated. Bottom: Conservation of the PP2A-B56-binding region in BRCA2. A Clustal Omega multiple sequence alignment of vertebrate BRCA2 protein sequences is shown with LxxIxE motif marked in blue and SQ/TQ sites in green. The sequence of the 2A mutation is shown. Asterisks (*) indicate conserved residues. Evolution tree using the TimeTree database[63] (timetree.org) is shown. MYA, million years ago. **c** Western blot of the co-immunoprecipitation of Myc-BRCA2[1001-1255] WT or 2A with Venus or Venus-B56γ from HeLa Flp-In T-REx cells in presence of 3 ng/mL Mitomycin C (MMC). Representative of three independent experiments. **d** Western blot of the co-immunoprecipitation of BRCA2 with B56 subunits from *Xenopus* egg extracts representative of two independent experiments. IP, immunoprecipitation. **e** Dissociation constants ($K_D$) for the interactions between the indicated BRCA2 peptides and B56α measured by isothermal titration calorimetry. **f** Western blot of a pulldown of an intact circular (Circ.) or linearized DSB-containing plasmid from mock (IgG) or BRCA2 immunodepleted *Xenopus* egg extracts representative of two independent experiments.

Expression of BRCA2 WT but to a lesser extent BRCA2 2A rescued the loss of RAD51 foci resulting from BRCA2 depletion (Fig. 2f). The impairment of RAD51 filament formation observed in the BRCA2 2A expressing cells did not arise from significant changes in the BRCA2-RAD51 interaction, as similar amounts of RAD51 co-purified with BRCA2 WT and 2A in immunoprecipitation assays (Fig. 2g).

Similar results were obtained when we deleted the entire conserved region, which contains the LxxIxE motif (BRCA2 Δ1100–1131). The BRCA2 Δ1100–1131 mutation caused a significant decrease in cell viability, DNA damage tolerance, and RAD51 foci formation (Supplementary Fig. 3a-e), in line with the results of the 2A mutation. We conclude that the interaction to PP2A-B56 is central to the function of BRCA2 in DNA repair by HR.

**ATM and ATR stimulate BRCA2-PP2A-B56 complex formation.** In several instances, PP2A-B56 interacts with substrate specifiers in a manner regulated by phosphorylation of neighboring sites flanking the LxxIxE motif to allow crosstalk between kinases and phosphatases[20]. The LxxIxE motif of BRCA2 is surrounded by three fully conserved SQ/TQ sites (Fig. 1b, Supplementary Data 1), which are putative consensus phosphorylation sites for the DNA damage response kinases[14]. To validate these phosphorylation sites, we raised phospho-specific

antibodies against the first and the last phosphorylation site, S1106 and T1128 (Fig. 3a, Supplementary Fig. 4a-b for antibody validation). For the S1106 phosphorylation site, the antigen included phosphorylation of T1104, which is a putative CDK site. We found that both phosphorylation of T1104/S1106 and T1128 are stimulated by CPT-induced DNA damage in S-phase (Fig. 3b). Inhibition of ATM and to a lesser extent ATR kinase reduced the phosphorylation, while inhibition of both fully abrogated it (Fig. 3b, Supplementary Fig. 4c). To dissect the kinetics of BRCA2 phosphorylation in a more synchronous model system, we turned to *Xenopus* egg extracts, taking advantage of the evolutionary conservation of the region surrounding T1128 (*X. laevis* T1196) (Fig. 3a), which allowed us to use the antibody raised against human BRCA2 pT1128. In this system, addition of a linearized DSB-containing plasmid, but not an intact circular one, resulted in rapid ATM-dependent BRCA2 T1196 phosphorylation (Supplementary Fig. 4d-e), which could also be detected on resected linearized DNA (Supplementary Fig. 4f). Likewise, during the replication-coupled repair of a cis-platin ICL containing plasmid[41], T1196 was phosphorylated at the time of DSB formation (Fig. 3c). Collectively, these results demonstrate that the SQ/TQ sites in BRCA2 flanking the LxxIxE motif are phosphorylated by ATM and ATR in response to DSBs.

Next, to directly assess whether phosphorylation of these sites affects the binding to PP2A-B56, we measured the binding

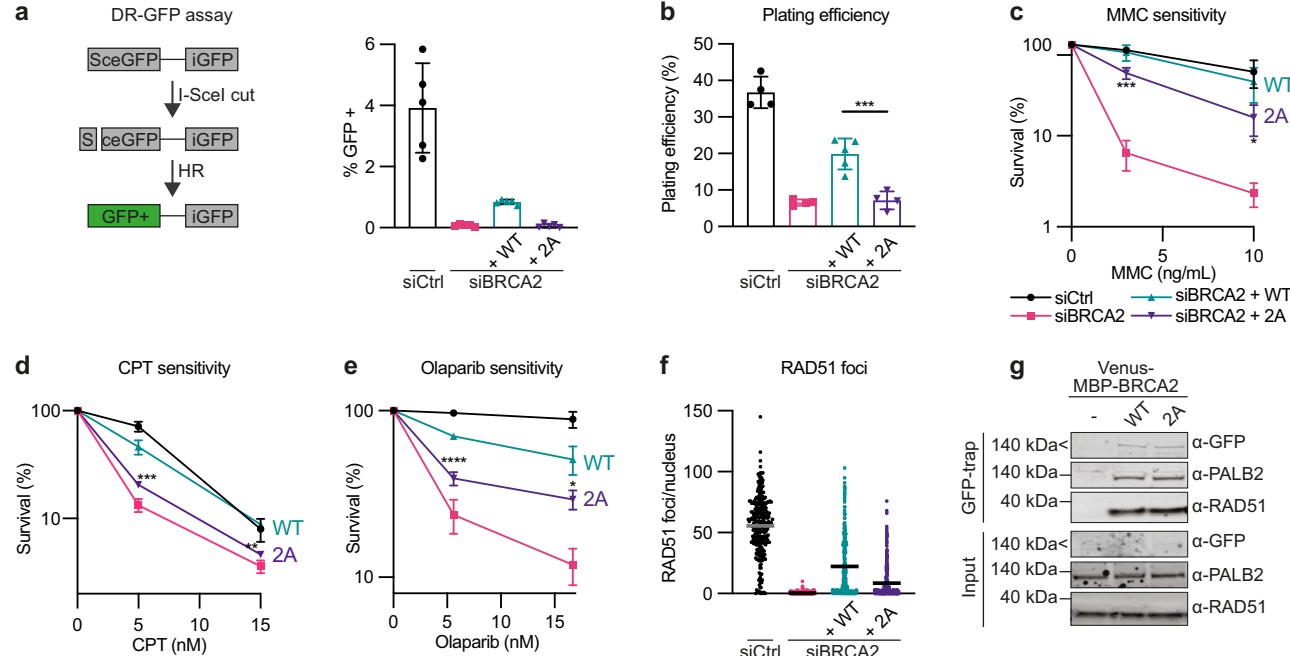

**Fig. 2 The BRCA2-PP2A-B56 complex is required for DNA repair by HR. a** Left: Schematic of the DR-GFP reporter assay. Right: Percentage of GFP positive (HR completed) HeLa DR-GFP Flp-In parental cells or stably expressing siRNA-resistant mCherry-MBP-BRCA2 WT or 2A cDNAs after transfection with Ctrl or BRCA2 siRNAs and an I-SceI-encoding plasmid, quantified by flow cytometry. Background values (without I-SceI) were subtracted. Data are presented as means ± standard deviations from five independent experiments, and individual data points are indicated. **b–f** U2OS Flp-In T-REx parental cells or stably expressing siRNA-resistant Venus-MBP-BRCA2 WT or 2A cDNAs were transfected with Ctrl or BRCA2 siRNAs. **b–e** Colony formation assays showing plating efficiency (**b**), MMC sensitivity (**c**), CPT sensitivity (**d**), and Olaparib sensitivity (**e**). Data are presented as means ± standard deviations for three independent experiments except for **b** and **c** for which siBRCA2 + WT is $n = 5$ and the remaining conditions are $n = 4$. One-way ANOVA analyses with Dunnett's multiple comparison tests were performed to compare each condition to siBRCA2 + WT. $*p < 0.5$, $**p < 0.1$, $***p < 0.001$, $****p < 0.0001$. **f** RAD51 nuclear foci in cells synchronized to S-phase with a thymidine block, released from the block and treated for 1 h with MMC, and then allowed to recover for 8 h before immunofluorescence microscopy. Each dot represents an individual nucleus, and means are indicated for 272 (siCtrl), 287 (siBRCA2), 239 (siBRCA2 + WT), or 297 (siBRCA2 + 2A) nuclei. The experiment is a representative of three independent experiments. **g** Western blot of the co-purification of RAD51 and PALB2 with Venus-MBP-BRCA2 WT and 2A from U2OS Flp-In T-REx cells representative of three independent experiments.

affinities between B56 and various phosphorylated BRCA2 peptides by ITC (Fig. 3d, Supplementary Fig. 5). Phosphorylation of S1123 and S1128 increased the binding affinity four and two fold, respectively, while the double phosphorylated peptide (S1123/S1128) had an eight-fold increase in binding affinity. In contrast, phosphorylation of S1106 slightly weakened the interaction.

To investigate how the phosphorylation status of BRCA2 affects PP2A-B56 binding in cells, we constructed mutants of BRCA2 with all three SQ/TQ sites mutated to AQ or DQ (referred to as BRCA2 3AQ and 3DQ), constituting unphosphorylated and phosphorylation-mimetic versions of the protein, respectively (Fig. 3a). We observed that Myc-BRCA2$^{1001–1255}$ 3AQ co-purified less with Venus-B56γ than Myc-BRCA2$^{1001–1255}$ WT, whereas Myc-BRCA2$^{1001–1255}$ 3DQ co-purified more with Venus-B56γ in immunoprecipitation assays (Fig. 3e), consistent with a two-fold increase in binding affinity of a 3DQ peptide measured by ITC (Fig. 3d). Our results argue that collectively these phosphorylations stimulate BRCA2 binding to PP2A-B56 in cells.

Next, to address whether these phosphorylation sites are important for the function of BRCA2, we investigated the viability, DNA damage tolerance and RAD51 focus formation of cells expressing BRCA2 3AQ and 3DQ in our RNAi and complementation system in U2OS Flp-In T-REx cells (Fig. 3f-j, Supplementary Fig. 2b-c). Expression of both BRCA2 3AQ and 3DQ resulted in decreased viability and MMC hypersensitivity

compared to BRCA2 WT (Fig. 3f-g), suggesting that dynamic phosphorylation of these sites is required for full BRCA2 functionality. However, while expression of BRCA2 3AQ led to CPT and Olaparib hypersensitivity and a reduction in RAD51 foci, BRCA2 3DQ was indistinguishable from BRCA2 WT in these assays (Fig. 3h-j). This indicates that BRCA2 3DQ, which can bind PP2A-B56 (Fig. 3d-e), supports some aspects of BRCA2 functionality including the ability to form RAD51 filaments at sites of DNA damage. In contrast, BRCA2 3AQ phenocopies BRCA2 2A (Fig. 2), consistent with its deficient PP2A-B56 binding (Fig. 3d-e).

Collectively, these results show that in response to DSBs, ATM, and ATR kinases phosphorylate conserved sites in BRCA2 flanking the LxxIxE motif, which in turn stimulate the formation of the BRCA2-PP2A-B56 complex.

**BRCA2 cancer mutations deregulate the PP2A-B56 interaction.** We next asked whether our findings would be clinically relevant to BRCA2 mutation carriers. Several BRCA2 missense variants of uncertain clinical significance, which are reported in individuals with a hereditary cancer predisposition, localize to the highly conserved B56-interacting region (ClinVar database, NIH). We selected three of them c.3318 C > G (S1106R), c.3346 A > C (T1116P), and c.3383 C > T (T1128I), which localize to the B56-regulating phosphorylation sites or the LxxIxE motif itself (Fig. 4a). Notably, BRCA2 S1106R was recently suggested to be likely benign using a multifactorial likelihood quantitative

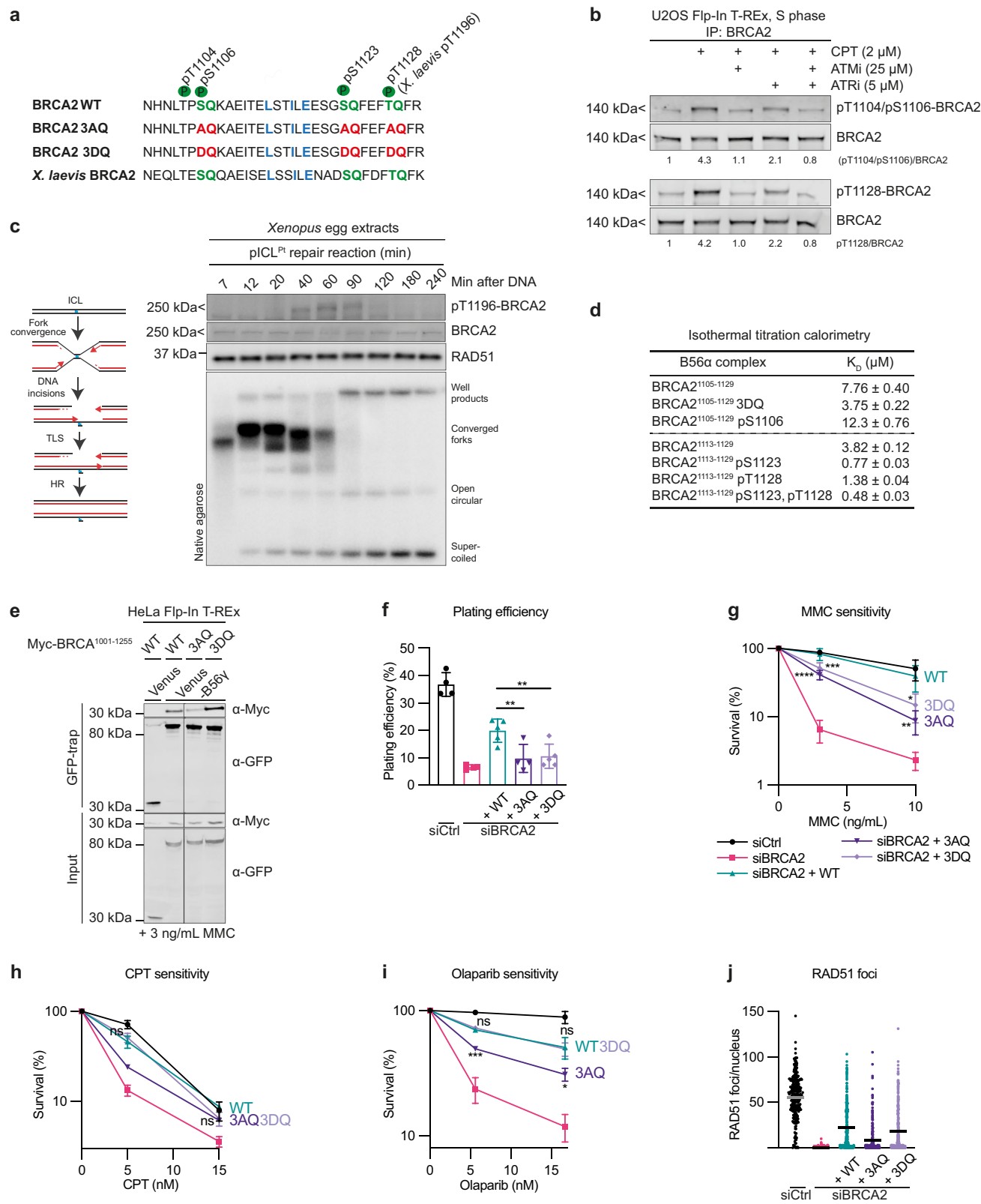

analysis[42]. We first determined whether these mutations interfere with PP2A-B56 binding. We observed that Myc-BRCA2[1001−1255] S1106R and T1116P co-purified more with Venus-B56γ than Myc-BRCA2[1001−1255] WT, whereas Myc-BRCA2[1001−1255] T1128I co-purified less with Venus-B56γ in immunoprecipitations assays (Fig. 4b). The increased binding of the S1106R mutant was reflected in a two-fold increase in binding affinity as determined

by ITC, whereas BRCA2 T1116P and T1128I had $K_D$ values similar to BRCA2 WT (Supplementary Fig. 6a-b). The stimulatory effect of S1106R likely arises from the generation of a positively charged motif upstream of the LxxIxE motif (Fig. 4a) that strengthen the binding of PP2A-B56[43]. T1116P generates a putative proline-directed phosphorylation site at position two of the LxxIxE motif (Fig. 4a), which is known to stimulate

**Fig. 3 ATM and ATR stimulate BRCA2-PP2A-B56 complex formation. a** Schematic of the conserved B56 binding region of human and *Xenopus laevis* BRCA2 with LxxIxE motif, relevant phosphorylation sites, and introduced mutations indicated. **b** Western blot analysis of BRCA2 immunoprecipitates from U2OS Flp-In T-REx cells, synchronized to S-phase by a thymidine block, released for 1 h, and then treated for 1 h with 2 μM CPT in presence or absence of ATM and ATR inhibitors. Representative of two independent experiments. The relative ratio of phosphorylated to total BRCA2 is indicated. **c** Left: simplified schematic of replication-coupled repair of a cisplatin ICL in *Xenopus* egg extracts. Right: pICL^Pt was replicated in *Xenopus* egg extracts and samples withdrawn at the indicated time points. Top: Western blot analysis. Bottom: Native agarose gel analysis of the same reaction supplemented with [α-32P] dATP. Representative of two independent experiments. **d** Dissociation constants ($K_D$) for the interactions between the indicated BRCA2 peptides and B56α measured by isothermal titration calorimetry. **e** Western blot of the co-immunoprecipitation of Myc-BRCA2^1001–1255 WT, 3AQ, or 3DQ with Venus-B56γ from HeLa Flp-In T-REx cells in presence of 3 ng/mL MMC. Representative of three independent experiments. The Myc-BRCA2^1001–1255 WT data (lanes 1–2) are identical to Fig. 1c, and the order of the lanes have been rearranged as indicated by black lines. **f–j** U2OS Flp-In T-REx parental cells or stably expressing siRNA-resistant Venus-MBP-BRCA2 WT, 3AQ, or 3DQ cDNAs were transfected with Ctrl or BRCA2 siRNA. The siCtrl, siBRCA2, and siBRCA2 + WT data are identical to Fig. 2b–f. **f–i** Colony formation assays showing plating efficiency (**f**), MMC sensitivity (**g**), CPT sensitivity (**h**), and Olaparib sensitivity (**i**). Data are presented as means ± standard deviations for three independent experiments except for in **f** and **g** where siBRCA2 + WT and siBRCA2 + 3DQ is $n = 5$ and the remaining conditions are $n = 4$. No statistical analysis is shown for the siBRCA2 + 3AQ condition in **h** as it is $n = 2$. One-way ANOVA analyses with Dunnett's multiple comparison tests were performed to compare each condition to siBRCA2 + WT. *$p < 0.5$, **$p < 0.1$, ***$p < 0.001$, ****$p < 0.0001$, ns nonsignificant. **j** RAD51 nuclear foci in cells synchronized to S-phase with a thymidine block, released from the block and treated for 1 h with MMC, and then allowed to recover for 8 h before immunofluorescence microscopy. Each dot represents an individual nucleus, and means are indicated for 272 (siCtrl), 287 (siBRCA2), 239 (siBRCA2 + WT), 278 (siBRCA2 + 3AQ), or 273 (siBRCA2 + 3DQ) individual nuclei. The experiment is a representative of three independent experiments.

interaction to PP2A-B56 when phosphorylated[20]. Finally, T1128I likely prevents the stimulatory effect of T1128 phosphorylation.

To address whether these cancer mutations impact on the function of BRCA2, we investigated the cell viability and DNA damage tolerance of cells expressing BRCA2 S1106R, T1116P, and T1128I in our RNAi and complementation system in U2OS Flp-In T-REx cells (Supplementary Fig. 6c). Expression of BRCA2 S1106R and T1128I resulted in diminished viability compared to expression of BRCA2 WT, and expression of all mutants led to a mild sensitivity to the clinically relevant PARP inhibitor Olaparib (Fig. 4c-d). Collectively, these results suggest that BRCA2 cancer mutations located in the B56-interacting region can deregulate the interaction to PP2A-B56 and sensitize cells to PARP inhibition.

## Discussion
Here, we uncover the protein phosphatase PP2A-B56 as a positive regulator of HR by the direct interaction to the HR component BRCA2. We propose a model (Fig. 4e) in which DSBs induce ATM/ATR-mediated phosphorylation of BRCA2 at S1106, S1123, and T1128. These phosphorylations stimulate the binding of PP2A-B56 to BRCA2 via a conserved LxxIxE motif, thus recruiting PP2A-B56 to the site of the lesion. The phosphorylation-regulated complex of BRCA2 and PP2A-B56 is required for efficient RAD51 filament formation and HR-mediated repair. This mechanism elegantly enables crosstalk between the DNA damage response and BRCA2-PP2A-B56 complex formation, possibly to ensure proper spatiotemporal formation of the complex.

A major question arising from our findings is what the functional substrate(s) of BRCA2-bound PP2A-B56 are at the site of the DNA lesion. Our results clearly illustrate that PP2A-B56 does not act as a mere off switch for DNA damage response signaling once repair is completed. Rather, the observation that the PP2A-B56 nonbinding mutant of BRCA2 is deficient in RAD51 filament formation and DNA repair by HR demonstrates that PP2A-B56 plays an active role during HR. BRCA2-bound PP2A-B56 may act to dephosphorylate protein substrates to positively moderate their functions in HR. It is also possible that BRCA2-bound PP2A-B56 is required for dynamic phosphorylation/dephosphorylation cycles of protein substrates at the site of the DNA lesion to drive RAD51 filament formation and repair. Based on our previous findings[29], we anticipate that the functional substrate(s) of BRCA2-bound PP2A-B56 will be found in close proximity to the

complex, and candidate substrates are proteins directly involved in RAD51 filament formation such as BRCA2 itself, RAD51 and RPA as well as BRCA1-PALB2-BRCA2 complex components PALB2 and BRCA1[4–6,12]. However, despite extensive efforts, we have not been able to identify the functionally relevant phosphorylation site(s) regulated by BRCA2-bound PP2A-B56.

In addition to HR, BRCA2 is involved in other processes such as fork protection and cohesin dynamics[44–46] and whether PP2A-B56 regulates BRCA2 function in these will be important to establish. Interestingly, during mitosis, PP2A-B56 appears to regulate BRCA2 function through an alternative recruitment mechanism[47], suggesting that PP2A-B56 might be a general regulator of BRCA2 functionality throughout the cell cycle.

Importantly, our discovery raises the possibility that mutations in PP2A-B56 components, which are common in human cancers[48], result in HR deficiencies that may be targeted therapeutically[39].

## Methods
**Cell culture**. HeLa cells (ATCC), U2OS Flp-In T-REx cells (a kind gift from Helen Piwnica-Worms), HeLa Flp-In T-REx cells (a kind gift from Stephen Taylor), HeLa DR-GFP Flp-In cells (a kind gift from Jeffrey Parvin), and derived cell lines from these were cultured in Dulbecco's Modified Eagle Medium with GlutaMAX (Life Technologies) supplemented with 10% fetal bovine serum (Gibco) and 10 units/mL of penicillin and 10 μg/mL of streptomycin (Gibco) at 37 °C with 5% CO2. Expression from the CMV-TetO2 promoter in Flp-In T-REx cells was induced by treatment with 10 ng/mL doxycycline (Clontech) for 24 h unless otherwise stated. To synchronize cells to S-phase, cells were incubated in growth medium with 2.5 mM thymidine (Sigma) for 24 h unless otherwise indicated. Cells were released from thymidine by washing twice in PBS and adding growth medium. Mitomycin C (MMC, Sigma), camptothecin (CPT, Sigma), Olaparib (AZD2281, Selleckchem), KU55933 (ATM kinase inhibitor, Selleckchem), and AZ20 (ATR kinase inhibitor, Selleckchem) were added at the indicated doses to the growth medium.

**Cloning**. A vector for stable high-level expression of BRCA2 in human cells, pcDNA5/FRT/hCMV/Venus-MBP-BRCA2, was generated by swapping the tetracycline-regulated CMV-TetO2 promoter in pcDNA5/FRT/TO with the high-level expression hCMV promoter from phCMV1 using MluI and BspTI restriction sites. To further increase the stability of BRCA2, Venus and MBP were inserted using HindIII and KpnI restriction sites. Finally, full-length BRCA2 was PCR amplified from pHA-BRCA2 (generous gift from Tina Thorslund) and inserted using KpnI and NotI restriction sites to generate pcDNA5/FRT/hCMV/Venus-MBP-BRCA2. To facilitate site-directed mutagenesis of full-length BRCA2, two cloning cassettes were generated using the internal NheI restriction site in combination with either KpnI or NotI encompassing BRCA2 CDS nucleotide positions 1–4584 and 4578–10257, respectively. These fragments were used as templates to introduce mutations in the PP2A-B56 binding region and silent mutations to obtain siRNA-resistance, respectively, and then reintroduced into pcDNA5/FRT/hCMV/Venus-MBP-BRCA2. For generation of pcDNA5/FRT/hCMV/mCherry-

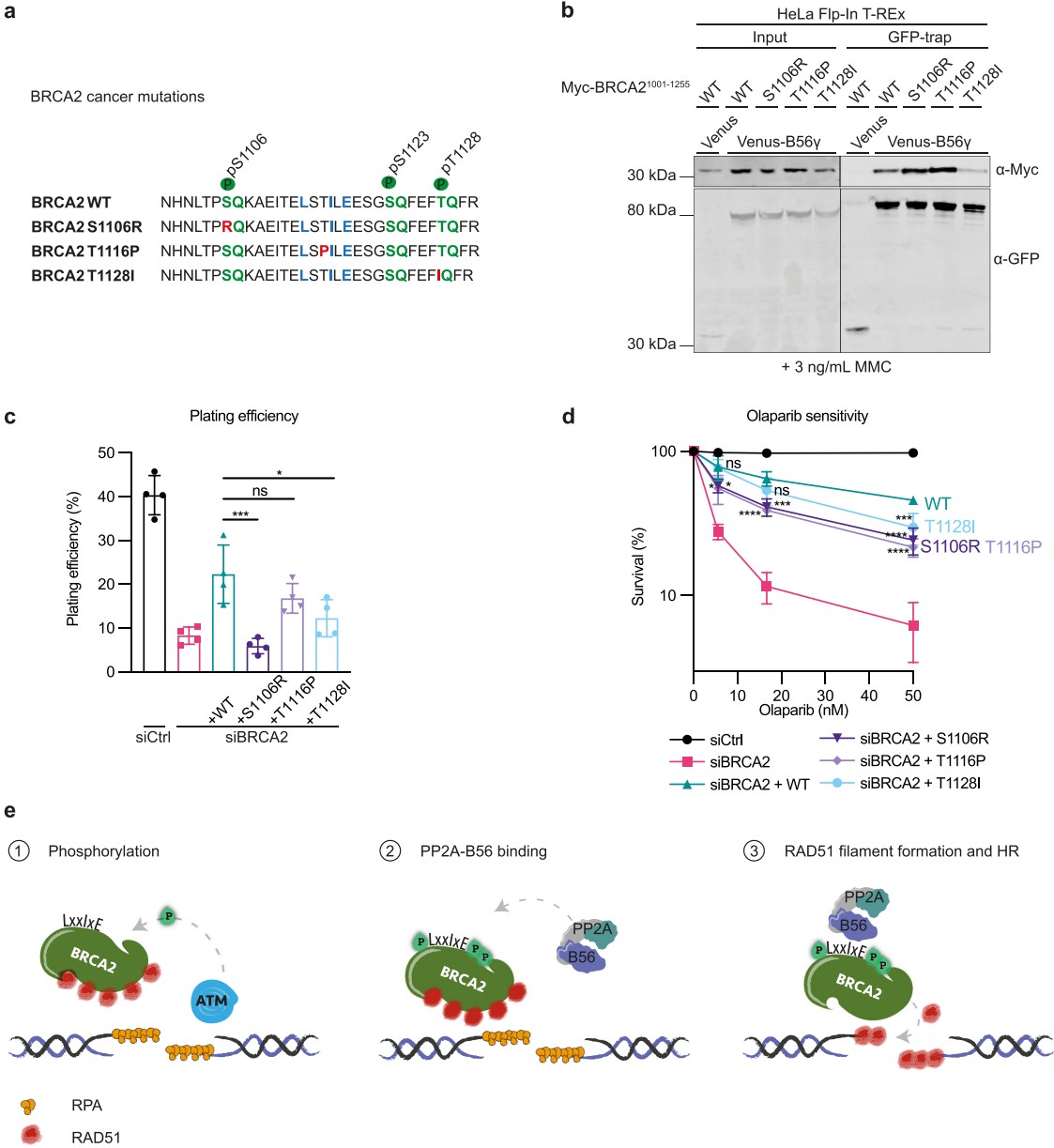

**Fig. 4 BRCA2 cancer mutations deregulate PP2A-B56 binding and sensitize cells to PARP inhibition. a** Schematic of the B56 binding region of human BRCA2 with the introduced cancer-associated mutations indicated. **b** Western blot of the co-immunoprecipitation of Myc-BRCA2[1001–1255] WT, S1106R, T1116P, or T1128I with Venus-B56γ from HeLa Flp-In T-REx cells in presence of 3 ng/mL MMC. Representative of three independent experiments. **c, d** U2OS Flp-In T-REx parental cells or stably expressing siRNA-resistant WT, S1106R, T1116P, or T1128I Venus-MBP-BRCA2 cDNAs were transfected with Ctrl or BRCA2 siRNA. Colony formation assays were performed to determine plating efficiency (**c**) and Olaparib sensitivity (**d**). Data are presented as means ± standard deviations for four independent experiments. One-way ANOVA analyses with Dunnett's multiple comparison tests were performed to compare each condition to siBRCA2 + WT. *$p < 0.5$, ***$p < 0.001$, ****$p < 0.0001$, ns nonsignificant. **e** Model. In the presence of DNA damage such as DSBs, ATM and ATR kinases phosphorylate BRCA2 on S1106, S1123, and T1128. This stimulates the binding of PP2A-B56 through a conserved LxxIxE motif, thus recruiting PP2A-B56 to the broken DNA. The phosphorylation-regulated binding of PP2A-B56 to BRCA2 is required for RAD51 filament formation and DNA repair by HR.

MBP-BRCA2, a synthetic cDNA of mCherry-MBP was synthesized (GeneArt) and swapped for Venus and MBP using HindIII and KpnI restriction sites. A vector for inducible expression of BRCA2 fragments in human cells for biochemistry, pcDNA5/FRT/TO/Myc-BRCA2[1001–1255], was generated by PCR amplifying BRCA2[1001–1255] with Myc tag-encoding overhangs and subsequent subcloning into pcDNA5/FRT/TO using BamHI and NotI restriction sites. Site-directed mutagenesis was performed to introduce mutations in the PP2A-B56 binding region. Similarly, pcDNA5/FRT/TO/3xFLAG-Venus-BRCA2[1001–1255] was generated by PCR amplification of BRCA2[1001–1255] and subsequent subcloning into pcDNA5/FRT/TO/3xFLAG-Venus using BamHI and NotI restriction sites. pcDNA5/FRT/TO/HA-TurboID-B56γ was generated by cloning B56γ into pcDNA5/FRT/TO/HA-TurboID. Primer sequences are enclosed in Supplementary Table 1.

Additionally, pcDNA5/FRT/TO/Venus-B56γ1[49], pcDNA5/FRT/TO/mCherry-B56 inhibitor, and pcDNA5/FRT/TO/mCherry-Ctrl inhibitor (3 A)[29] were used in this study.

**Generation of stable Flp-In T-REx cell lines.** U2OS Flp-In T-REx, HeLa Flp-In T-REx, or HeLa DR-GFP Flp-In cells were grown in medium supplemented with 100 μg/mL Zeocin (Invitrogen). To generate stable cell lines in the Flp-In system, cells were co-transfected with pOG44 (Invitrogen) and a pcDNA5/FRT plasmid of interest using the Fugene 6 transfection kit (Promega) or Lipofectamine 2000 (Invitrogen). After transfection, Flp-In cells were selected in medium supplemented with 200 μg/mL Hygromycin B (Invitrogen). Individual clones were selected and

analyzed for expression. For T-REx cells, selection included 5 µg/mL blasticidin S HCl (Sigma).

**Transfection**. For transient protein expression, cells were transfected with Lipofectamine 2000 (Invitrogen) and the plasmid of interest and incubated for 48 h unless otherwise stated. For BRCA2 knockdown, cells were transfected twice with 10 nM Silencer Select BRCA2 s2084 siRNA and 10 nM Silencer Select BRCA2 s2085 siRNA (Ambion) using Lipofectamine RNAiMAX (Invitrogen) 24 and 48 h before the experiment. A luciferase oligo (5′-CGUACGCGGAAUA CUUCGAdTdT-3′, Sigma) was used for control (Ctrl).

**DR-GFP reporter assay**. To analyze HR efficiency for full-length BRCA2 constructs, HeLa DR-GFP Flp-In parental cells or stably expressing siRNA-resistant mCherry-MBP-BRCA2 were transfected with Ctrl or BRCA2 siRNA as described above. The second siRNA transfection was combined with transient transfection with or without an I-SceI-encoding plasmid. After 48 h, cells were trypsinized, dissolved in 2% BSA in PBS, stained with 1 µg/mL DAPI, and analyzed on a BD LSRFortessa flow cytometer (BD biosciences) using the BD FACSDiva software (version 9, BD biosciences). FSC (A, W, H), SSC (A), DAPI (A), and GFP (A) were acquired. The data were analyzed in FCS express (version 7.04, De Novo Software). Debris and doublets were excluded by gating. Living cells were gated by excluding DAPI positive cells. The fraction of GFP positive cells was quantified (example of gating strategy is shown in Supplementary Fig. 2d) and the background (without I-SceI endonuclease) was subtracted for each condition. Graphs were constructed in PRISM (version 9.1.1., GraphPad). For the B56 inhibitor experiment, HeLa DR-GFP Flp-In cells were transiently transfected with a plasmid encoding an mCherry-tagged version of the B56 substrate inhibitor or a control version of the inhibitor described previously[29] either with or without an I-SceI-encoding plasmid. After 48 h, cells were prepared and analyzed as described above but using mCherry (A) to gate transfected cells. The fraction of GFP positive cells in the mCherry positive population was quantified (example of gating strategy is shown in Supplementary Fig. 2f), and the background (without I-SceI endonuclease) was subtracted for each condition. Graphs were constructed in PRISM, and a two-tailed Student's t-test was performed to determine the p value.

**Colony formation assay**. U2OS Flp-In T-REx parental cells or stably expressing siRNA-resistant venus-MBP-BRCA2 constructs were transfected with Ctrl or BRCA2 siRNA as described above. Then, cells were either treated with 0, 3, or 10 ng/mL Mitomycin C for 24 h followed by reseeding into normal growth medium or reseeded directly and either treated for 24 h with 0, 5, or 15 nM CPT or continuously maintained in medium containing 0, 5.6, 16.7, or 50 nM Olaparib. Reseeding was performed by trypsinizing the cells, dissolving into growth medium, and counting the number of cells using the Scepter Cell Counter (Merck), followed by seeding a known number of cells into six-well plates containing growth medium. After 11 days, the cells were fixed and stained in 0.5% methylviolet, 25% methanol. The plates were scanned on a GelCount (Oxford Optronix), and the number of colonies were quantified using the GelCount software (version 1.3.04). The plating efficiency (%) for each well was calculated as the number of colonies divided by the number of cells seeded times 100. The surviving fraction for each dose of drug was calculated by normalizing the plating efficiency to that of the unperturbed condition. The plating efficiency data of Figs. 2, 3, and Supplementary Fig. 3 is derived from the unperturbed condition of the MMC sensitivity dataset, and the plating efficiency data of Fig. 4 is derived from the unperturbed condition of the Olaparib sensitivity dataset. Graphs were constructed in PRISM, and one-way ANOVA analyses with Dunnett's multiple comparison tests were performed comparing the mean of each condition to the siBRCA2 + WT condition. Exact p values are provided in the source data.

**Immunofluorescence microscopy**. U2OS Flp-In T-REx parental cells or stably expressing siRNA-resistant venus-MBP-BRCA2 constructs were seeded in µ-Slide eight-well dishes (Ibidi). Alongside Ctrl or BRCA2 siRNA transfection as described above, cells were synchronized to S-phase with a single 24 h 2 mM thymidine block. Cells were released from the block, treated with 3 µM MMC for 1 h, and then allowed to recover for 8 h in normal growth medium. Cells were fixed and permeabilized by incubation in 4% formaldehyde for 10 min, 0.1% Triton-X-100 in PBS-T for 10 min, and 25 mM glycine for 20 min, followed by blocking in 3% BSA (Sigma) in PBS-T for 30 min. Cells were incubated with primary antibody, rabbit-anti-RAD51 (Bioacademia 70-001) 1:1000 in blocking solution, for 90 min, followed by washing in TBS-T and incubation with secondary antibody, AlexaFluor 546 nm Goat-anti-rabbit IgG (Life Technologies, A-11010) 1:1000 and 1 µg/mL DAPI, in blocking solution for 45 min. Finally, cells were washed in PBS-T and analysed on a Deltavision Elite microscope with a ×40 oil objective using Delta-Vision SoftWoRx software (version 7.0.0., GE healthcare). Images were deconvoluted using DeltaVision SoftWoRx (version 5.0.0., GE healthcare), and Z stacks combined using the Quick projection function. The number of RAD51 foci in each nucleus was quantified using the polygon finder function. Graphs were constructed in PRISM.

**Antibodies**. Commercially available antibodies against the following proteins were used for western blotting in the indicated dilutions: BRCA2 (Millipore Cat# OP95, RRID:AB_2067762, 1:1000), RAD51 (Bio Academia Cat# 70-001, 1:1000), mCherry (RFP) (MBL International Cat# PM005, RRID:AB_591279, 1:1000), Myc (Santa Cruz Biotechnology Cat# sc-40, RRID:AB_627268, 1:750), PALB2 (Bethyl Cat# A301-246A, RRID:AB_890607, 1:1000), GAPDH (Santa Cruz Biotechnology Cat# sc-25778, RRID:AB_10167668, 1:5000), Tubulin (Abcam Cat# ab6160, RRID:AB_305328, 1:5000), Histone 3 (Abcam Cat# ab1791, RRID:AB_302613, 1:1000), pS345-CHK1 (Cell Signaling Technology Cat# 2341, RRID:AB_330023, 1:1000), pS1981-ATM (Millipore Cat# MAB3806, RRID:AB_569379, 1:2000), PP2A-C (Millipore Cat# 05-421, RRID:AB_309726, 1:1000). Additionally, an antibody against GFP was used (Serum produced by Moravian, affinity purified against full-length GFP) at dilution 1:5000. Phospho-specific polyclonal antibodies against pT1104/pS1106-BRCA2 and pT1128-BRCA2 were raised in rabbits using phosphorylated peptides of BRCA2 for immunization, affinity purification, and validation (SNHNL(pT)P(pS)QKAEI for pT1104/pS1106-BRCA2 (21st Century Biochemicals) and CQFEF(pT)QFRKPS for pT1128-BRCA2 (Moravian)) and were used for western blotting at dilution 1:500.

Antibodies against Xenopus MCM6[50] (1:2500), BRCA2[51] (1:5000), BRCA2[52] (Supplementary Fig. 4d; 1:500), RAD51[53] (1:2500), RPA70[54] (1:2500), and ORC2[55] (1:2500) were described previously. Additional antibodies against the following Xenopus proteins were raised in rabbits against the following peptides: BRCA2 (Ac-KPHIKEDQNEPESNSEYC-amide, New England Peptide) as described previously[52] for the purpose of BRCA2 immunodepletions, WRN (H2N-MTSLQRKLPEWMSVKC-amide, New England Peptide, 1:2500), B56α/β (MSAISAAEKVDGFTRKSVRK, Peptide Speciality Laboratories GmbH, 1:1000), and B56γ (MPNKNKKDKEPPKAGKSGKS, Peptide Speciality Laboratories GmbH, 1:500). The antibody against Xenopus pT1196-BRCA2 was raised against human pT1128-BRCA2 (see above).

**Whole-cell extracts, immunoprecipitation, and western blotting**. For whole-cell extracts, cells were lysed in ice-cold RIPA buffer (10 mM Tris, pH 7.4, 150 mM NaCl, 1 mM EDTA, 1% NP-40, 0.5% sodium deoxycholate, 0.1% SDS), and cell lysates were cleared by centrifugation at $20,000 \times g$ at 4 °C. Protein concentrations in cell lysates were determined using the Bradford protein assay kit (Bio-Rad) or Pierce BCA protein assay kit (Thermo Fisher Scientific).

For GFP-trap immunoprecipitation of Venus and Venus-B56γ, HeLa Flp-In T-REx cells stably expressing doxycycline-inducible Venus or Venus-B56γ were transiently transfected with the indicated constructs of pcDNA5/FRT/TO/Myc-BRCA2[1001–1255], induced with 10 ng/mL doxycycline, and incubated with 3 ng/mL MMC for 24 h prior to cell harvest. Cells were lysed in ice-cold low salt lysis buffer (50 mM Tris, pH 7.4, 50 mM NaCl, 1 mM EDTA, 0.1% Igepal). Cell lysates were cleared by centrifugation at $20,000 \times g$ at 4 °C, and proteins were purified by GFP-trap (ChromoTek) immunoprecipitation for 1 h at 4 °C. Beads were washed in ice-cold no salt wash buffer (50 mM Tris pH 7.4, 20% glycerol, 1 mg/mL BSA) prior to elution.

For GFP-trap immunoprecipitation of Venus and Venus-BRCA2[1001–1255], HeLa cells were transiently transfected with pcDNA5/FRT/TO/Venus or pcDNA5/FRT/TO/Venus-BRCA2[1001–1255], synchronized to S-phase as described above, released for 2 h and then treated for 2 h with 100 nM CPT prior to cell harvest. Cells were lysed and proteins purified by GFP-trap immunoprecipitation in low salt lysis buffer as described above. Beads were washed in low salt lysis buffer prior to elution.

For GFP-trap immunoprecipitation of Venus-MBP-BRCA2, U2OS Flp-In T-REx stably expressing constructs of Venus-MBP-BRCA2 were lysed and proteins immunoprecipitated as described above but in a standard salt lysis buffer (50 mM Tris, pH 7.4, 150 mM NaCl, 1 mM EDTA, 0.1% Igepal).

For immunoprecipitations of endogenous BRCA2, U2OS Flp-In T-REx cells were synchronized to S-phase as described above, released for 1 h, and treated for 1 h with 2 µM CPT in presence or absence of 25 µM KU55933 (ATM kinase inhibitor) and 5 µM AZ20 (ATR kinase inhibitor). Cells were lysed in RIPA buffer, and proteins were immunoprecipitated on BRCA2 antibody-conjugated (Millipore Cat# OP95, RRID:AB_2067762) Rec-protein G Sepharose 4B beads (Invitrogen) for 1 h at 4 °C and washed in RIPA buffer prior to elution.

All buffers were supplemented with 1 mM DTT, Complete protease inhibitor cocktail (Roche), and PhosSTOP phosphatase inhibitor cocktail (Roche). For λ phosphatase treatment experiments, immunoprecipitants on beads were washed in buffer without phosphatase inhibitor and incubated with λ phosphatase (Sigma–Aldrich) in the applied buffer for 20 min at 30 °C before elution. Immunoprecipitants were eluted in 2× NuPage LDS sample buffer (Invitrogen). Whole-cell extracts and immunoprecipitations were analyzed by SDS-PAGE and western blotting or mass spectrometry analysis. For western blotting, samples were boiled for 5 min in NuPage LDS sample buffer and run on NuPage Bis-Tris 4–12% protein gels (Invitrogen), and proteins were transferred to PVDF membranes (Immobilon-FL, Merck). For dot blots, the indicated peptides were spotted onto nitrocellulose membranes (Hybond-C extra, Amersham Biosciences) in five-fold dilutions (highest amount 2 µg). Xenopus samples (see below) were prepared in 2× Laemmli sample buffer, boiled for 5 min, run on 4–12% Criterion XT Bis-Tris Protein Gels (Bio-rad), and proteins were transferred to Polyscreen (R) PVDF transfer membranes (PerkinElmer). All membranes were blocked in 5% skim milk

or BSA, incubated in primary antibody solution overnight at 4 °C, washed in TBS-T, incubated in secondary antibody for 1 h, washed again in TBS-T, and imaged with the Odyssey® CLx (LI-COR) using ImageStudio (version 3.1.4., LI-COR) or incubated with ECL reagent and imaged on an ImageQuant LAS4000 (Cytiva) using ImageQuant LAS4000 software (version 1.2., GE healthcare). Quantification of western blots were carried out in ImageStudioLite (version 5.2.5., LI-COR). Uncropped western blot images are shown in Supplementary Fig. 7.

**Fractionation assay.** U2OS Flp-In T-REx cells stably expressing Venus-MBP-BRCA2 were transfected with BRCA2 siRNA as described above prior to lysis in low salt lysis buffer. Upon clearing of the lysates, supernatants were stored as the soluble fractions. The pellets were resuspended and lysed in RIPA buffer supplemented with benzonase nuclease (Merck Millipore). Lysates were centrifuged again, and the supernatants were stored as the chromatin fractions. The soluble and chromatin fractions were analyzed by SDS-PAGE and western blotting.

**Biotin proximity labeling assay.** HeLa Flp-In T-REx encoding doxycycline-inducible TurboID-B56γ were induced with 4 ng/mL doxycycline alongside synchronization to S-phase as described above. Cells were released for 2 h in presence of 100 nM CPT, and 50 μM biotin (Sigma) was added 30 min before harvest. Biotinylated proteins were purified on High Capacity Streptavidin Agarose beads (Thermo Scientific) in RIPA buffer and proteins were identified by mass spectrometry.

**Protein expression and purification.** BRCA2[1089–1143] WT and 2A (L1114A-I1117A) were cloned into pGEX-4T-1 to generate N-terminally GST-tagged fusion proteins. Constructs were transformed into *E. coli* BL21 (DE3) cells and expression was induced by addition of 0.5 mM IPTG at 37 °C for 3 h. Bacterial pellets were resuspended in ice-cold lysis buffer (50 mM Tris-HCl pH 7.4, 300 mM NaCl, 10% glycerol, 5 mM β-mercaptoethanol, 1 mM phenylmethyl sulfonyl fluoride, and complete EDTA-free Protease Inhibitor Cocktail tablets (Roche)) and lysed in an EmulsiFlex-C3 High Pressure Homogenizer (Avestin). Lysates were cleared at 26,200 × *g* for 30 min at 4 °C and supernatants were incubated with pre-washed Glutathione Sepharose 4 Fast Flow beads (GE Healthcare) for 90 min at 4 °C with mixing. Beads were washed six times in ice-cold lysis buffer, and GST-fusion proteins were eluted at 22 °C for 30 min, 1250 rpm in elution buffer (50 mM Tris pH 8.8, 300 mM NaCl, 10% glycerol, 5 mM β-mercaptoethanol, 20 mM reduced glutathione). Eluates were further purified by gel filtration on a Superdex 75 10/300 GL column.

His-tagged B56α was expressed in the *E. coli* strain BL21 Rosetta2 (DE3) R3 T1 at 18 °C for 20 h using 0.5 mM IPTG. The bacterial pellets were resuspended in ice-cold buffer L (50 mM NaP, 300 mM NaCl, 10% Glycerol, 0.5 mM TCEP, pH 7.5) containing complete EDTA-free Protease Inhibitor Cocktail tablets and lysed with an EmulsiFlex-C3 High Pressure Homogenizer. The lysate was centrifuged at 18,500 × *g* for 30 min and the supernatant filtered through a 0.22 μm PES filter and loaded onto a 1 mL Ni column (GE healthcare) in buffer L with 10 mM immidazole, washed and eluted. The eluate was loaded on a Superdex 200 PG 16/60 equilibrated with SEC buffer (50 mM NaP, 150 mM NaCl, 0.5 mM TCEP, 10% Glycerol, pH 7.50) and fractions were analyzed by SDS-PAGE and verified by mass spectrometry.

Biotinylated LacI was expressed and purified as previously described[56]. Briefly, pET11a-LacI and pBirAcm (Avidy) were co-transformed in T7 Express Competent Cells (NEB) and cultured in the the presence of 100 mg/mL ampicillin and 34 mg/mL chloramphenicol at 37 °C until OD600 reached ~0.6. The culture was then supplemented with 1 mM IPTG and 50 mM biotin for 2 h. Cells were then pelleted by centrifugation, resuspended in lysis Buffer 1 (50 mM Tris pH 7.5, 5 mM EDTA, 100 mM NaCl, 10% sucrose, 1 mM DTT, protease inhibitors (Roche), 0.2 mg/mL lysozyme (Sigma), and 0.1% Brij 58) and rotated for 30 min at room temperature. The lysate was then pelleted by high-speed centrifugation (60 min at 20,000 × *g*), and the pellet was resuspended in 10 mL of Extraction Buffer (50 mM Tris pH 7.5, 5 mM EDTA, 1 M NaCl, 30 mM IPTG, 1 mM DTT, and protease inhibitors). The resuspension was homogenized by sonication and pelleted again (60 min at 20,000 × *g*). The supernatant was collected and 1% polymin P was added to 0.045%. The lysate was then rotated for 30 min at 4 °C and pelleted again (20 min at 20,000 × *g*). The supernatant was collected, transferred to a new tube and ammonium sulfate was added to it to a final saturation of 37% followed by rotation for 30 min at 4 °C. The pellet was recovered by centrifugation and resuspended in 2 mL of Wash Buffer (50 mM Tris pH 7.5, 1 mM EDTA, 100 mM NaCl, 1 mM DTT, and protease inhibitors). The resuspension was applied to a column containing 1 mL of softlink avidin resin and rotated for 1 h at 4 °C. The column was washed with 15 mL of Wash Buffer, and the protein eluted with Elution buffer (50 mM Tris pH 7.5, 1 mM EDTA, 100 mM NaCl, 1 mM DTT, and 5 mM biotin). The protein was then dialyzed in Dialysis Buffer (50 mM Tris pH 7.5, 1 mM EDTA, 150 mM NaCl, 1 mM DTT, and 30% glycerol) at 4 °C. The protein was recovered and stored at −80 °C at a concentration of ~1 mg/mL.

**Isothermal titration calorimetry (ITC).** Peptides were purchased from Peptide 2.0 Inc. (Chantilly, VA, USA). The purity obtained in the synthesis was 95–98% as determined by high performance liquid chromatography (HPLC) and subsequent

analysis by mass spectrometry. Both recombinant B56α and synthetic BRCA2 peptides were extensively dialyzed prior to ITC experiments against the ITC buffer (50 mM sodium phosphate pH 7.5, 150 mM NaCl, 0.5 mM TCEP). All experiments were performed on a MicroCal Auto-iTC200 (Malvern Panalytical) instrument at 25 °C (Auto-iTC200 version 1.1.1.0, iTC200 version 1.26.4, Origin version 7.0552 SR4). Both peptide and B56α concentrations were determined using a spectrophotometer by measuring the absorbance at 280 nm and applying values for the extinction coefficients as computed from the corresponding sequences by the ProtParam program (http://web.expasy.org/protparam/). The BRCA2 peptides were loaded into the syringe and titrated into the calorimetric cell containing B56α. The reference cell was filled with distilled water. Control experiments with the peptides injected in the sample cell filled with buffer were carried out under the same experimental conditions. These control experiments showed negligible heats of dilution in all cases. The titration sequence consisted of a single 0.4 μl injection followed by 19 injections, 2 μl each, with 150 s spacing between injections to ensure that the thermal power returns to the baseline before the next injection. The stirring speed was 750 rpm. The heats per injection normalized per mole of injectant versus the molar ratio [BRCA2 peptide]/[B56α] were fitted to a single-site model. Data were analysed with MicroCal PEAQ-ITC analysis software (version 1.1.0.1262., Malvern Panalytical). All ITC data including Gibbs free energy (ΔG), enthalpy (ΔH), entropy (−TΔS), equilibrium dissociation constant (KD) and reaction stoichiometry (n) are shown in Supplementary Table 2.

**Gel filtration.** To analyze the binding between BRCA2 and B56α by gel filtration, 100 μg of B56α was incubated with 40 μg of GST or GST-BRCA2[1089–1143] in buffer G (150 mM NaCl, 25 mM Tris 8.0, 10% glycerol, 1 mM DTT) in a total volume of 525 μl. Following incubation, the sample was loaded on a Superdex 200 10/300 column (GE Healthcare) and fractions were analysed by SDS-PAGE and Coomassie blue staining.

**Label-free LC-MS/MS analysis.** Pulldowns were analyzed on a Q-Exactive Plus quadrupole or Fusion Orbitrap Lumos mass spectrometer (ThermoScientific) equipped with Easy-nLC 1000 or 12000 (ThermoScientific) and nanospray source (ThermoScientific). Peptides were resuspended in 5% methanol/1% formic acid and loaded onto a trap column [1 cm length, 100 μm inner diameter, ReproSil, C18 AQ 5 μm 120 Å pore (Dr. Maisch, Ammerbuch, Germany)] vented to waste via a micro-tee and eluted across a fritless analytical resolving column (35 cm length, 100 μm inner diameter, ReproSil, C18 AQ 3 μm 120 Å pore) pulled in-house (Sutter P-2000, Sutter Instruments, San Francisco, CA) with a 45 min gradient of 5–30% LC-MS buffer B (LC-MS buffer A: 0.0625% formic acid, 3% ACN; LC-MS buffer B: 0.0625% formic acid, 95% ACN). The Q-Exactive Plus was set to perform an Orbitrap MS1 scan (R = 70 K; AGC target = 1e6) from 350 to 1500 *m/z*, followed by HCD MS2 spectra on the 10 most abundant precursor ions detected by Orbitrap scanning (R = 17.5 K; AGC target = 1e5; max ion time = 50 ms) before repeating the cycle. Precursor ions were isolated for HCD by quadrupole isolation at width = 1 *m/z* and HCD fragmentation at 26 normalized collision energy (NCE). Charge state 2, 3, and 4 ions were selected for MS2. Precursor ions were added to a dynamic exclusion list ±20 ppm for 15 s.

Raw data were searched using COMET (release version 2014.01) in high resolution mode[57] against a target-decoy (reversed)[58] version of the human proteome sequence database (UniProt; downloaded 2/2020, 40704 entries of forward and reverse protein sequences) with a precursor mass tolerance of ±1 Da and a fragment ion mass tolerance of 0.02 Da, and requiring fully tryptic peptides (K, R; not preceding P) with up to three mis-cleavages. Static modifications included carbamidomethylcysteine and variable modifications included: oxidized methionine and STY phosphorylation. Searches were filtered using orthogonal measures including mass measurement accuracy (±3 ppm), Xcorr for charges from +2 through +4, and dCn targeting a <1% FDR at the peptide level. Quantification of LC-MS/MS spectra was performed using MassChroQ[59] and the iBAQ method[60]. Missing values were imputed from a normal distribution in Perseus (version 1.6.14.0., MaxQuant) to enable statistical analysis[61]. For further analysis, proteins had to be identified in the B56γ + dox +biotin or Venus-BRCA2 samples with more than 1 total peptide and quantified in 2 or more replicates. B56γ or BRCA2 protein abundances were normalized to be equal across all samples. Statistical analysis was carried out in Perseus by two-tailed Student's *t*-test.

***Xenopus* egg extract preparation and reaction.** *Xenopus* egg extracts were prepared as described before[62] using *Xenopus laevis* (Nasco Cat #LM0053MX, LM00715MX). All experiments involving animals were approved by the Danish Animal Experiments Inspectorate and conform to relevant regulatory standards and European guidelines.

For replication of pICL[Pt], the plasmid was first licensed in high-speed supernatant (HSS) extract for 30 min at RT at a final DNA concentration of 7.5 ng/μL. DNA replication was then initiated by adding two volumes of nucleoplasmic egg extract (NPE). For all other nonreplicating reactions, DNA was supplemented to NPE at a final concentration of 15 ng/μL. When indicated, ATM inhibitor (KU55933, Selleckchem), ATR inhibitor (AZ20, Sigma), or DNA-PK inhibitor (NU 7441, Selleckchem) were added to NPE to a final concentration of 100 μM 10 min prior to initiating the reaction. To visualize DNA replication intermediates, reactions were

supplemented with [α-$^{32}$P] dATP (Perkin Elmer) and 1.5 μL of each time point was added to 5 μL of stop buffer (5% SDS, 80 mM Tris pH 8.0, 0.13% phosphoric acid, 10% Ficoll). Proteins were digested by adding 1 μL of Proteinase K (20 mg/mL) (Roche) for 1 h at 37 °C. Replication intermediates were separated by 0.9% native agarose gel electrophoresis and visualized using a phosphorimager.

DNA constructs used in *Xenopus* egg extract experiments pICL$^{Pt}$ was prepared as previously described[41]. To generate closed circular or linear DNA substrates, pBlueScript was either untreated or linearized with XhoI and purified via a DNA purification spin column (Qiagen).

**Immunoprecipitations and immunodepletions from *Xenopus* egg extracts.** To immunodeplete BRCA2 from NPE, one volume of Protein A Sepharose Fast Flow (PAS) (GE Health Care) beads was bound to five volumes of affinity purified BRCA2 antibody (1 mg/mL) overnight at 4 °C. The beads were then washed once with PBS, once with ELB (10 mM HEPES pH 7.7, 50 mM KCl, 2.5 mM MgCl$_2$, and 250 mM sucrose), twice with ELB supplemented with 0.5 M NaCl, and twice with ELB. One volume of NPE was then depleted by mixing with 0.2 volumes of antibody-bound beads incubated at room temperature for 15 min. The supernatant was recovered, and the depletion procedure repeated three additional times. The mock depletion was performed similarly using purified IgG from pre-immune serum. For immunoprecipitation experiments, 5 μL of PAS beads were incubated with 10 μg of the indicated affinity purified antibody. The sepharose beads were washed twice with PBS and three times with IP buffer 1 (10 mM Hepes pH 7.7, 50 mM KCl, 2.5 mM MgCl$_2$, 0.25% NP-40). Five microliters of NPE was diluted with 20 μL of IP buffer and incubated with antibody prebound beads for 1 h at RT. The beads were then washed three times with IP buffer and resuspended in 50 μL of 2× Laemmli sample buffer before analysis by western blotting.

**Plasmid pulldown from *Xenopus* egg extracts.** For plasmid pulldown experiments, 10 μL of streptavidin-coupled magnetic beads (Dynabead M-280, Invitrogen) per pulldown reaction were equilibrated with wash buffer 1 (50 mM Tris-HCl, pH 7.5, 150 mM NaCl, 1 mM EDTA pH 8, 0.02% Tween 20) and then incubated with 12 pmol of biotinylated LacI at RT for 40 min. The beads were washed four times with pulldown buffer (10 mM Hepes pH 7.7, 50 mM KCl, 2.5 mM MgCl$_2$, 250 mM sucrose, 0.02% Tween 20). 225 nanograms of either closed circular or linear pBlueScript was bound to beads for 45 min. The beads were then washed twice with pull-down buffer and resuspended in 15 μL of NPE supplemented with Tween 20 to a final concentration of 0.02%. The reaction was incubated for 15 min at RT, washed twice in pull-down buffer and resuspended in 30 μL of 2× Laemmli sample buffer before analysis by western blotting.

**Reporting summary**. Further information on research design is available in the Nature Research Reporting Summary linked to this article.

## Data availability

The data that support this study are available from the corresponding authors upon reasonable request. Vertebrate BRCA2 protein sequences used for Clustal Omega multiple sequence alignment were downloaded from the NCBI protein database (https://www.ncbi.nlm.nih.gov/protein/). Evolution tree was generated using the TimeTree database (timetree.org). The human proteome sequence database used for mass spectrometry analysis were downloaded (2/2020) from UniProt (https://www.uniprot.org). The mass spectrometry data generated in this work have been deposited to ProteomeXchange under accession code PXD027574 and MassIVE under accession code MSV000087884. The DR-GFP reporter assay data, colony survival assay data, and RAD51 foci data generated in this study are provided in the source data. Source data are provided with this paper.

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

## Acknowledgements

Work at the Novo Nordisk Foundation Center for Protein Research is supported by grant NNF14CC0001 and J.N.I. is supported by grants from the Danish Cancer Society (R167-A10951-17-S2), Independent Research Fund Denmark (8021-00101B) and Novo Nordisk Foundation (NNF18OC0053124 and NNF20OC0065098). This work was furthermore supported by the Danish Cancer Society (R146-A9454-16-S2) to M.L. and the Villum Foundation to V.H.O. and M.L., the Danish National Research Foundation (DNRF115) to M.L., Dansk Kræftforskningsfond to S.M.A., and by the R35GM119455 grant from the National Institute of General Medicine to A.N.K. We thank the CPR protein production facility for helping to produce and purify recombinant B56 protein and the DanStem Flow Cytometry and Sequencing (FlowCytSeq) platform for technical assistance with flow cytometry. We would also like to thank Helen Piwnica-Worms, Stephen Taylor, and Jeffrey Parvin for the kind gifts of the U2OS Flp-In T-REx, HeLa Flp-In T-REx, and HeLa DR-GFP Flp-In cell lines, respectively. Additionally, we thank Tina Thorslund for sharing the pHA-BRCA2 cDNA with us. Furthermore, we want to thank Johannes C. Walter for sharing *Xenopus* antibodies and reagents as well as Vicenzo Costanzo for sharing the *Xenopus* BRCA2 antibody. The PP2A-B56-LxxIxE motif model used to generate Fig. 1a was kindly provided by Rebecca Page.

## Author contributions

S.M.A. performed all experimental work with the following exceptions. J.P.D. performed the *Xenopus* egg extract experiments. E.P.T.H. performed the conservation analysis. E.P.T.H., T.K., and V.H.O. contributed to cloning and establishment of the RNAi and complementation setup and generated preliminary data. I.N., L.E.C., and A.N.K. performed the mass spectrometry experiments. J.D. performed the TurboID experiment and made the artwork for the model in Fig. 4e. B.L.M. generated the ITC data. B.R. generated the *Xenopus* B56 antibodies. T.v.O.H. gave clinical input on BRCA2 patient mutations. E.P.T.H. and J.N. purified recombinant proteins, and J.N. performed gel filtration experiments. V.H.O., M.L., and J.N. supervised the project. S.M.A. drafted the manuscript. All authors contributed to the writing of the manuscript.

## Competing interests

J.N. is on the scientific advisory board for Orion Pharma. T.v.O.H. has received lecture honoraria from Pfizer. The remaining authors declare no competing interests.
