## [Peer Review File · Nature Communications]

A complex of BRCA2 and PP2A-B56 is required for DNA repair by homologous recombinationREVIEWER COMMENTS

Reviewer #1 (Remarks to the Author):

- What are the noteworthy results?

For me this work opened new perspectives in understanding such a general process as regulation of protein function by reversible phosphorylation and in particular in understanding the mechanism of action of BRCA2 in DNA repair and homologous recombination. At least since 1992 when Fischer and Krebs received Nobel Prize

for their discoveries of reversible protein phosphorylation as a biological regulatory mechanism, we know that adding or removing phosphate groups to proteins by action of specific protein kinases or phosphatases can work as activating or inactivating switches. Ambjoern et al., demonstrate a novel interplay between DNA damage signaling kinases acting on BRCA2 protein and specific protein phosphatase PP2A-B56, which binds to phosphorylated BRCA2 protein and is having there an architectural role needed for proper functioning of BRCA2 during DNA repair.

- Will the work be of significance to the field and related fields? How does it compare to the established literature? If the work is not original, please provide relevant references.

It is a new concept that in my opinion agrees very well with what we knew before but brings our understanding to a higher level.

- Does the work support the conclusions and claims, or is additional evidence needed?

The panoply of complex and perfectly designed experiments supported beyond any reasonable doubt all conclusions and claims of the authors.

- Are there any flaws in the data analysis, interpretation and conclusions? - Do these prohibit publication or require revision?

I did not see any flaws in the MS.

- Is the methodology sound? Does the work meet the expected standards in your field?

Yes, the methodology is of very high quality.

- Is there enough detail provided in the methods for the work to be reproduced?

Yes.

I just would have one suggestion for a correction. On page 3 the authors wrote: "BRCA2 plays a central role in HR by facilitating the formation of RAD51 nucleoprotein filaments on resected RPA-coated single-stranded DNA ends."

I would replace "facilitating " by "controlling" since as long as the cell is not "ready" for DNA repair, BRCA2 binds RAD51 and apparently prevents it from binding to ssDNA. See for example: Role of BRCA2 in control of the RAD51 recombination and DNA repair protein. Molecular Cell, 2001, 273-282.

Reviewer #2 (Remarks to the Author):

The manuscript by Ambjoern et al, A complex of BRCA2 and PP2A-B56 is required for DNA repair by homologous recombination reveals a previously unknown role for PP2A-B56 in DNA repair via

homologous recombination in a rigorous and convincing manner. Authors demonstrate that BRCA2 contains a functional B56 binding motif of the LxxIxE type that is required for DNA repair and is regulated during the repair process via phosphorylation of flanking residues that increases the affinity of BRCA2 for PP2A-B56 and results in increased binding. Finally, authors examine some patient derived mutations in the relevant regions of BRCA2 and show that they indeed alter binding to PP2A. The study is strengthened by a number of quantitative assays that convincingly reveal effects of mutations that alter the LxxIxE motif in BRCA2 and/or the phosphorylated flanking residues. The figures are clearly presented and the experiments are all appropriately controlled.

Thus in total the work establishes a new regulatory mechanism that is clearly required for efficient DNA repair and shows the importance of identifying phosphatase substrates using specific short linear motifs. Until recently this was the key limitation in protein phosphatase research and this work exemplifies why the contributions of protein phosphatases need to be elucidated. PP2A-B56 is an essential regulator of this process—a very important finding.

The only disappointment, as noted by the authors, is the failure to identify one or more substrates for PP2A-B56. There is minimal discussion of this, or even what might be key candidate substrates. It seems obvious that RAD51 whose phosphorylation is known to regulate its recruitment to sites of damage and function in DNA repair is one such candidate. Was this tested? Even inclusion of negative results, i.e. "we tested X candidate substrates but were unsuccessful in demonstrating dephosphorylation" would be a significant addition to the manuscript.

Signed by Martha S Cyert

Reviewer #3 (Remarks to the Author):

In this manuscript by Nilsson and colleagues a describe a novel interaction between protein phosphatase PP2A-B56 and BRCA2 in mediating homologous recombination. The authors show that BRCA2 is phosphorylated by ATM and lesser so ATR within a highly conserved LxxIxE motif between the BRC1 and BRC2 repeats of BRCA2. These are also binding sites of PP2A-B56 to BRCA with then the authors suggest control the HR process. Furthermore, the authors suggest that the certain BRCA2 cancer associated mutations could deregulate the binding of BRCA2 and PP2A thus deregulating the HR process.

This is a very interesting paper elucidating a novel role of BRCA2-PP2A control in homologous recombination process. The biochemical experiments using human cells and *Xenopus* egg extracts are very clear delineating the control of DSB repair by HR. However the phenotypic experiments do not completely recapitulate the biochemical experiments and my suggestion is that the authors need to work with cleaner genetic systems to elucidate the physiological relevance of this interaction.

Major comments

1. In figure 2A the authors use DR-GFP assays to quantify the level of HR upon siRNA with BRCA2 and then transiently expressing WT BRCA2 and 2A mutant. The authors are doing multiple transfections in this experiment. How are they controlling the transfection efficiency of BRCA2 and 2A mutant in this experiment. As it is the efficiency of this experiment tends to be very low (4-6%) after normalizations. Furthermore, the expression of BRCA2 WT in siBRCA2 seems to be quite low as the rescue of HR is at best 15-20%. Therefore, it is very difficult to assess the efficiency of 2A mutant in this scenario. I would suggest the authors use stable inducible overexpression systems to control their experiments better.

2. In the clonogenic assays, why are the 2A mutants have an intermediate phenotype when compared with siBRCA2 in MMC and Olaparib sensitivity? If this interaction is so crucial, my understanding is that it should phenocopy siBRCA2. Furthermore the same clonogenic experiments

need to be performed with siBRCA2 and the mutant overexpression showing the epistasis.

3. For RAD51 foci experiment in 2F, the transfection efficiency seems to be an issue again where overexpression of WT BRCA2 is only rescuing the effects slightly and it would be difficult to assess the role of 2A mutant.

4. The authors observe in Fig2 G that the interaction between BRCA2 and Rad51 is not disrupted by 2A mutant. Since BRCA2 is the loader of RAD51 at DSBs, how do the authors envision the RAD51 loading defect in 2A mutants that they claim to observe.

5. The authors should also either perform genetic depletion of PP2A and/or inhibit PP2A to test the epistasis with BRCA2 in terms of sensitivity to MMC and Olaparib.

6. Again to better understand the effect of the cancer-associated mutations, the authors should use an inducible system and a stable expression system in BRCA2 siRNA or BRCA2 null tumor cells to assess the effects of these mutations on cellular viability to strengthen their claims.

Minor Comments:

Replace the BRCA2 blot in Fig.S2A with a cleaner blot if possible.

Our response to the reviewers' comments (in bold).

Reviewer #1 (Remarks to the Author):

- What are the noteworthy results?

For me this work opened new perspectives in understanding such a general process as regulation of protein function by reversible phosphorylation and in particular in understanding the mechanism of action of BRCA2 in DNA repair and homologous recombination. At least since 1992 when Fischer and Krebs received Nobel Prize for their discoveries of reversible protein phosphorylation as a biological regulatory mechanism, we know that adding or removing phosphate groups to proteins by action of specific protein kinases or phosphatases can work as activating or inactivating switches. Ambjoern et al., demonstrate a novel interplay between DNA damage signaling kinases acting on BRCA2 protein and specific protein phosphatase PP2A-B56, which binds to phosphorylated BRCA2 protein and is having there an architectural role needed for proper functioning of BRCA2 during DNA repair.

Our response: We thank the reviewer for the positive comments on our work.

- Will the work be of significance to the field and related fields? How does it compare to the established literature? If the work is not original, please provide relevant references.

It is a new concept that in my opinion agrees very well with what we knew before but brings our understanding to a higher level.

- Does the work support the conclusions and claims, or is additional evidence needed?

The panoply of complex and perfectly designed experiments supported beyond any reasonable doubt all conclusions and claims of the authors.

- Are there any flaws in the data analysis, interpretation and conclusions? - Do these prohibit publication or require revision?

I did not see any flaws in the MS.

- Is the methodology sound? Does the work meet the expected standards in your field?

Yes, the methodology is of very high quality.

- Is there enough detail provided in the methods for the work to be reproduced?

Yes.

I just would have one suggestion for a correction. On page 3 the authors wrote: "BRCA2 plays a central role in HR by facilitating the formation of RAD51 nucleoprotein filaments on resected RPA-coated single-stranded DNA ends."

I would replace "facilitating" by "controlling" since as long as the cell is not "ready" for DNA repair, BRCA2 binds RAD51 and apparently prevents it from binding to ssDNA. See

for example: Role of BRCA2 in control of the RAD51 recombination and DNA repair protein. *Molecular Cell*, 2001, 273-282.

Our response: We have changed the wording according to this suggestion.

Reviewer #2 (Remarks to the Author):

The manuscript by Ambjoern et al, A complex of BRCA2 and PP2A-B56 is required for DNA repair by homologous recombination reveals a previously unknown role for PP2A-B56 in DNA repair via homologous recombination in a rigorous and convincing manner. Authors demonstrate that BRCA2 contains a functional B56 binding motif of the LxxIxE type that is required for DNA repair and is regulated during the repair process via phosphorylation of flanking residues that increases the affinity of BRCA2 for PP2A-B56 and results in increased binding. Finally, authors examine some patient derived mutations in the relevant regions of BRCA2 and show that they indeed alter binding to PP2A. The study is strengthened by a number of quantitative assays that convincingly reveal effects of mutations that alter the LxxIxE motif in BRCA2 and/or the phosphorylated flanking residues. The figures are clearly presented and the experiments are all appropriately controlled.

Our response: We thank the reviewer for the positive comments on our work.

Thus in total the work establishes a new regulatory mechanism that is clearly required for efficient DNA repair and shows the importance of identifying phosphatase substrates using specific short linear motifs. Until recently this was the key limitation in protein phosphatase research and this work exemplifies why the contributions of protein phosphatases need to be elucidated. PP2A-B56 is an essential regulator of this process-a very important finding.

The only disappointment, as noted by the authors, is the failure to identify one or more substrates for PP2A-B56. There is minimal discussion of this, or even what might be key candidate substrates. It seems obvious that RAD51 whose phosphorylation is known to regulate its recruitment to sites of damage and function in DNA repair is one such candidate. Was this tested? Even inclusion of negative results, i.e. "we tested X candidate substrates but were unsuccessful in demonstrating dephosphorylation" would be a significant addition to the manuscript.

Our response:

We agree with the reviewer that it would be informative to identify the functional substrates of PP2A-B56 bound to BRCA2. Based on our current knowledge (Kruse et al., 2020), we anticipate that the substrates will be in close proximity to PP2A-B56-BRCA2 and therefore likely candidates are proteins directly involved in RAD51 loading such as BRCA2, RAD51 and RPA as well as BRCA1-PALB2-BRCA2 complex components PALB2 and BRCA1 (Jensen et al., 2010; Liu et al., 2010; Thorslund et al., 2010; Xia et al., 2006). In the revised discussion, we have speculated on what the relevant substrates could be.

Despite extensive efforts we have not been able to establish the functional relevant phosphorylation site(s) regulated by PP2A-B56-BRCA2. Our efforts include:

- 1) Phosphoproteomic analysis of affinity purified Venus-MBP-BRCA2 WT vs. Venus-MBP-BRCA2 2A. This did not result in clear and consistent results.**
- 2) Use of specific phosphoantibodies comparing BRCA2 WT vs BRCA2 mutant conditions or using our genetically encoded inhibitor of substrate binding to LxxIxE motifs. These attempts included phosphoantibodies (commercially available or our own generated for this study) against BRCA2 pS3291, BRCA2 pT1104/S1106, BRCA1 pS1387, BRCA1 pS1423, and RPA32 (phosphoshifts). These approaches did not result in clear and consistent results.**
- 3) In vitro dephosphorylation of BRCA2 phosphopeptides with either WT or 2A LxxIxE motifs. This showed that S1106 is efficiently dephosphorylated by PP2A-B56 in an LxxIxE motif-dependent manner. T1128 is also a PP2A-B56 substrate but less dependent on the LxxIxE motif. Based on these results, we analysed whether S1106 could be a functionally relevant phosphorylation site. However, we detected no phenotype of BRCA2 S1106A or BRCA2 S1106D in colony formation assays and therefore concluded that this is not the case.**

Since these efforts did not result in conclusive results and failed to identify functional relevant phosphorylation sites in BRCA2 regulated by PP2A-B56, we did not include them in the manuscript. In our view, identifying and characterizing the relevant substrates and phosphorylation sites will be an important future endeavor.

Signed by Martha S Cyert

Reviewer #3 (Remarks to the Author):

In this manuscript by Nilsson and colleagues a describe a novel interaction between protein phosphatase PP2A-B56 and BRCA2 in mediating homologous recombination. The authors show that BRCA2 is phosphorylated by ATM and lesser so ATR within a highly conserved LxxIxE motif between the BRC1 and BRC2 repeats of BRCA2. These are also binding sites of PP2A-B56 to BRCA with then the authors suggest control the HR process. Furthermore, the authors suggest that the certain BRCA2 cancer associated mutations could deregulate the binding of BRCA2 and PP2A thus deregulating the HR process.

This is a very interesting paper elucidating a novel role of BRCA2-PP2A control in homologous recombination process. The biochemical experiments using human cells and Xenopus egg extracts are very clear delineating the control of DSB repair by HR. However the phenotypic experiments do not completely recapitulate the biochemical experiments and my suggestion is that the authors need to work with cleaner genetic systems to elucidate the physiological relevance of this interaction.

Our response:

We thank the reviewer for the positive comments on our manuscript. We want to point out that we have used isogenic stable cell lines generated by use of the FRT Flp-in system throughout our work and never transiently transfected BRCA2 constructs (see details below).

Major comments

1. In figure 2A the authors use DR-GFP assays to quantify the level of HR upon siRNA with BRCA2 and then transiently expressing WT BRCA2 and 2A mutant. The authors are doing multiple transfections in this experiment. How are they controlling the transfection efficiency of BRCA2 and 2A mutant in this experiment. As it is the efficiency of this experiment tends to be very low (4-6%) after normalizations. Furthermore, the expression of BRCA2 WT in siBRCA2 seems to be quite low as the rescue of HR is at best 15-20%. Therefore, it is very difficult to assess the efficiency of 2A mutant in this scenario. I would suggest the authors use stable inducible overexpression systems to control their experiments better.

Our response:

We want to point out that the reviewer did not fully understand our experimental setup. In the text we write:

“To address this, we constructed an RNAi knockdown and complementation set-up in HeLa DR-GFP Flp-In cells²⁵ and U2OS Flp-In T-REx cells. This setup allowed transient depletion of endogenous BRCA2 using siRNA-mediated knockdown and complementation with stably expressed siRNA-resistant cDNA constructs of mCherry- or Venus-MBP-tagged full-length BRCA2 WT or 2A (referred to as BRCA2 WT and 2A).”

To clarify: We have used cell lines that contain a FRT site allowing us to stably integrate mCherry-MBP-BRCA2 WT or 2A expression cassettes. These cell lines were analysed to show that equal levels of mCherry-MBP-BRCA2 was expressed (Western blot in Supplementary Fig. 2a). Therefore, BRCA2 is stably expressed and there is no transient transfection with BRCA2 constructs. We have thus already done the experiment the way the reviewer requests.

It is correct when the reviewer states that we do not achieve a full rescue of the BRCA2 RNAi phenotype. This is true both for this experiment and in the other experiments in the manuscript. We suspect that this is due to the Venus-MBP/mCherry-MBP-tagged BRCA2 transgene being less active than the endogenous BRCA2. To achieve detectable expression of full-length BRCA2 it was necessary to tag it with Venus-MBP/mCherry-MBP and these tags could interfere with the function. Throughout the manuscript we therefore compare tagged BRCA2 variants to the tagged BRCA2 wild type and make conclusions based on this comparison.

With regards to the level of recombination observed in the DR-GFP assay, we report 2-6% GFP-positive cells after transfection of the I-SceI nuclease, which is similar to the level reported by other labs (e.g. Pierce et al., 1999).

2. In the clonogenic assays, why are the 2A mutants have an intermediate phenotype

when compared with siBRCA2 in MMC and Olaparib sensitivity? If this interaction is so crucial, my understanding is that it should phenocopy siBRCA2.

Our response:

In the MMC and Olaparib colony formation assays we have a larger phenotypic window compared to the DR-GFP reporter assay in part due to more efficient complementation of the BRCA2 RNAi by Venus-MBP-BRCA2 WT in this setup. Therefore, we were able to detect an intermediate phenotype of BRCA2 2A in these experiments. We do not anticipate and do not claim that BRCA2 2A is a complete null. In our view, multiple functional domains of BRCA2 collectively contributes to functionality and removing one of these will compromise function but not completely abolish it.

Furthermore the same clonogenic experiments need to be performed with siBRCA2 and the mutant overexpression showing the epistasis.

Our response:

The experiments have been done with siBRCA2. We always use a parental cell line treated with a control RNAi oligo or treated with BRCA2 RNAi oligoes as controls. All stable cell lines expressing Venus-MBP-BRCA2 are treated with BRCA2 RNAi oligoes to remove endogenous BRCA2 allowing us to compare phenotypes of wild type vs. mutants.

3. For RAD51 foci experiment in 2F, the transfection efficiency seems to an issue again where overexpression of WT BRCA2 is only rescuing the effects slight bit and it would be difficult to assess the role of 2A mutant.

Our response:

As stated above, we use stable cell lines and the phenotype of cells expressing the BRCA2 2A transgene is compared to cells expressing the BRCA2 WT transgene.

4. The authors observe in Fig2 G that the interaction between BRCA2 and Rad51 is not disrupted by 2A mutant. Since BRCA2 is the loader of RAD51 at DSBs, who do the authors envision the RAD51 loading defect in 2A mutants that they claim to observe.

Our response:

See also our comments to reviewer 2 on substrates of BRCA2-PP2A-B56. We anticipate that proteins in proximity of BRCA2-PP2A-B56 will be substrates of PP2A-B56-mediated dephosphorylation. Likely candidates include proteins directly involved in RAD51 loading such as BRCA2, RAD51 and RPA as well as BRCA1-PALB2-BRCA2 complex components PALB2 and BRCA1 (Jensen et al., 2010; Liu et al., 2010; Thorslund et al., 2010; Xia et al., 2006). The dephosphorylation of these substrates may potentially facilitate efficient RAD51 filament formation through activation of an HR component or by allowing dynamic loading of RAD51. To establish this, we need to identify the phosphorylation sites regulated by BRCA2-PP2A-B56, which we believe is beyond the scope of the current study.

In the revised discussion we have speculated on these scenarios.

5. The authors should also either perform genetic depletion of PP2A and/or inhibit PP2A to test the epistasis with BRCA2 in terms of sensitivity to MMC and Olaparib.

Our response:

As PP2A is essential we cannot perform long term genetic depletion/inhibition. Instead, we have used our genetically encoded PP2A-B56 inhibitor that blocks the binding of PP2A-B56 to LxxIxE motifs. We have tried to use this inhibitor in clonogenic survival assays to look at sensitivity to Olaparib. Unfortunately, the long-term inhibition of PP2A-B56 is strongly antiproliferative making the inhibitor impossible to use in clonogenic survival assays.

Instead, we have used this inhibitor in the more short-term DR-GFP reporter assays, and here, we observed an inhibition of HR when we blocked the binding of PP2A-B56 to LxxIxE motifs (Supplementary Fig. 2d).

6. Again to better understand the effect of the cancer associate mutations, the authors should use an inducible system and a stable expression system in BRCA2 siRNA or BRCA2 null tumor cells to assess the effects of these mutations on cellular viability to strengthen their claims.

Our response:

See also comments from above. All experiments are done with isogenic stable cell lines expressing Venus-MBP-BRCA2 variants and with knockdown of endogenous BRCA2. The requested experiments are therefore already included in the manuscript.

Minor Comments:

Replace the BRCA2 blot in Fig.S2A with a cleaner blot if possible.

Our response:

This is the best blot out of two.

References:

Kruse, T. et al. Mechanisms of site-specific dephosphorylation and kinase opposition imposed by PP2A regulatory subunits. *EMBO J.* **39**, e103695 (2020).

Jensen, R. B., Carreira, A. & Kowalczykowski, S. C. Purified human BRCA2 stimulates RAD51-mediated recombination. *Nature* **467**, 678–83 (2010).

Liu, J., Doty, T., Gibson, B. & Heyer, W. D. Human BRCA2 protein promotes RAD51 filament formation on RPA-covered single-stranded DNA. *Nat Struct Mol Biol* **17**, 1260–2 (2010).

Thorslund, T. et al. The breast cancer tumor suppressor BRCA2 promotes the specific targeting of RAD51 to single-stranded DNA. *Nat Struct Mol Biol* **17**, 1263–5 (2010).

Xia, B. et al. Control of BRCA2 cellular and clinical functions by a nuclear partner, PALB2. *Mol Cell* **22**, 719–29 (2006).

Pierce, A. J., Johnson, R. D., Thompson, L. H. & Jasin, M. XRCC3 promotes homology-directed repair of DNA damage in mammalian cells. *Genes Dev* **13**, 2633–8 (1999).